# Talent management of international nurses in healthcare settings: A systematic review

**Sidra Hareem Zulfiqar**[1]*, **Nuala Ryan**[2], **Elaine Berkery**[2], **Claire Odonnell**[3], **Helen Purtil**[4], **Bernadette O'Malley**[5]

**1** Department of Work and Employment Studies, Kemmy Business School, University of Limerick, Limerick, Ireland, **2** Department of Management and Marketing, Kemmy Business School, University of Limerick, Limerick, Ireland, **3** Department of Nursing Studies and Midwifery, School of Medicine, University of Limerick, Limerick, Ireland, **4** Department of Mathematics and Statistics, Science and Engineering, University of Limerick, Limerick, Ireland, **5** University Limerick Hospital Group, Limerick, Ireland

* Sidra.Zulfiqar@ul.ie

**Data Availability Statement:** All relevant data are contained within the paper and its Supporting Information files.

**Funding:** This research is funded by the British Academy of Management (see: 2021-Nuala Ryan

## Abstract

### Aim

To identify and systematically review current scholarship on talent management of international nurses in healthcare organizations.

### Background

As nurse shortages persistently pose challenges for healthcare organizations globally, one of the primary strategies employed to address these shortages is employment of international nurses. To date little has been done to systematically review and collate contemporary research on talent management of this strategically important cohort. Talent management is a holistic construct that can support healthcare organizations to attract, develop, motivate, and retain talented employees to drive organizational performance. This systematic review isolates, appraises and collates available evidence on talent management practices for international nurses.

### Study design

Systematic literature review.

### Data sources

Searches of PubMed, EBSCO and Scopus were made covering literature from 2012–2022.

### Review methods

This study followed Cochrane protocol for Systematic Reviews and key search terms were developed in consultation with University of Limerick library. As a key aim of the review was to provide evidence for the development of effective talent management practices, only peer-reviewed academic papers and empirical studies were included. Initial articles screening was conducted by two reviewers and full articles review was conducted by the entire research team. Findings were combined in a data extraction template for further analysis.

(bam.ac.uk)). (NR and EB) https://www.bam.ac.uk/grants/project-repository/funded-transitions-1-projects/2021-nuala-ryan.html Funding has also been received from the Martha McMenamin Memorial Scholarship (BO'M). Both funders of this study have no role in the study design, the collection and analysis of the data or the writing of the manuscript.

**Competing interests:** No authors have competing interests

## Results

This review includes 62 articles thematically analysed under the headings recruitment and selection, retention and turnover, career progression, professional development, discrimination and racism, culture and communication.

## Conclusion

No articles were found that directly address talent management for international nurses. Although there are studies that address aspects of talent management independently, more research is required on talent management as a holistic process for international nurses to inform evidence-based practice.

## Impact

This research emphasizes the importance of talent management for retention of international nurses in healthcare settings. It provides a knowledge base for healthcare organisations to enhance employee retention and ensure quality care for patients, as well as setting the foundation for future studies in this area.

## 1. Introduction

Globally there is an increasing demand for healthcare services due to ageing populations [1, 2], the prevalence of chronic diseases, and emerging global health challenges [3, 4]. This places additional pressure on the nursing workforce as healthcare organisations around the world are challenged with lack of staff [2, 4, 5] pressures exacerbated by the onset of Covid-19 pandemic [6–8]. In addition, many countries are facing an ageing nursing workforce, with a significant proportion of nurses nearing retirement [9] which creates space for new cohorts of nurses to replace those who are leaving. WHO 'estimates a projected shortfall of 10 million health workers by 2030' [10, p.1].

International healthcare staff are being employed by healthcare organisations in many countries as a solution to staff shortages [1, 11–14]. Addressing staff shortages requires organisations to adopt talent management practices to attract and retain competent staff [2, 11, 15]. Talent management can be understood as 'the process of attraction, development, and preservation of working people of increased abilities, skills, and knowledge' [16, p.1]. Studies on talent management in the field of nursing demonstrate that effective talent management is mutually beneficial for nurses, healthcare organisations and the wider healthcare system [1, 15–17].

Similarly, nurses' professional skills, job satisfaction and overall organisation's productivity increased manifold with efficient talent management practices [2, 16]. Whilst hiring international staff brings many cultural and economic benefits [8, 18], it comes with multiple challenges [1, 13, 18, 19] and ethical considerations [5, 8]. Research findings in [20] highlighted some of the key challenges that have worsened staff retention and intensified nurses' intention to leave; i.e., excessive workload [4, 18], stress and burnout [6], aging nursing workforce [9], migration and integration challenges [1, 5, 11, 13, 21], and especially the psychological, mental and emotional impact of Covid-19 [6, 8, 20]. Also, Covid-19 brought an overwhelming number of resignations by healthcare staff [6, 7] having serious implications on healthcare systems worldwide. Therefore, as nurses retire or choose to leave the profession, healthcare organisations continue to struggle in hiring both temporary and permanent staff [9, 20].

Similarly, the recruitment of international nurses from developing countries may have economic and social implications which affects the sustainability of international nurses' migration [11, 19] and causes serious issues of staff retention [2]. The ability of healthcare organisations to retain international nurses using clear and comprehensive available evidence on effective talent management practices must be a priority.

## 2. Background

With the growing need for organisations to develop and retain their employees, talent management, as a field of study, has been the focus of scholarship for the last two decades as it provides a knowledge base related to staff retention [2, 11, 17, 21]. The lack of staff retention has serious implications on providing quality healthcare services and patient care [11, 17] thus, talent management is beginning to gain attention in the healthcare sector to address the challenges of staff turnover and intention to leave [1, 2, 11, 17]. Therefore, it is more important than ever to systematically collate, understand and evaluate the existing literature on the topic of talent management for internationally trained nurses in healthcare organizations to inform an evidenced based approach to these practices. This type of review can also present a narrative around the importance of talent management practices for internationally trained nurses in healthcare organizations, as well as highlighting directions for further research.

## 3. The review

### 3.1 Aim and objectives

The aim of this paper is to identify and evaluate current scholarship on talent management of international nurses and to systematically review the identified publications on the topic. This paper aims to meet the following objectives which is consistent with the protocol paper for this study [12]:

1. Systematically identify and review the existing research on talent management for internationally trained nurses in healthcare settings based on the study protocol,

2. Identify the key themes from the literature,

3. Identify the gaps existing in current literature,

4. Present the findings from the review to provide evidence around the development of talent management practices for healthcare organisations.

### 3.2 Study design

The stepwise process of systematic review was conducted using online searches on medical and nursing related databases. As the first step, before formally starting the search process, the pilot searches were run on multiple databases like CINAHL with full text (EBSCOhost), PubMED, PsycINFO, Embase, Business Source Complete, Academic Source Complete, Web of Science, and Medline, as outlined in the studies protocol [12]. Preliminary searches were run with generic search terms like international nurses, talent management, migrant nurses, etc. to make sure that relevant studies can be found in the available literature.

### 3.3 Search methods

Based on consultations with the relevant librarians in the Glucksman Library at the University of Limerick the following search terms included 'internationally trained nurses' or 'foreign

nurses' or 'migrant nurses' and 'talent management', 'attraction', recruitment', 'retention', 'development', 'succession planning', 'leadership', 'performance management'. The Boolean term 'and' was used to combine the main term of 'international nurses or overseas nurses or foreign nurses or migrant nurses' with each of the following individual terms one by one. The individual terms to be combined with main term in separate searches are described as follows:

- talent management

- attraction

- recruitment and selection

- career progression or promotion or career advancement or career development or career trajectories

- talent identification

- career management

- learning and development

- promotions in the workplace

- succession planning and leadership development

- retention or attrition or turnover or intent to leave or intent to stay

    Resulting in the following 10 search strings:

1. 'international nurses or overseas nurses or foreign nurses or migrant nurses' AND 'talent management'

2. 'international nurses or overseas nurses or foreign nurses or migrant nurses' AND 'career management'

3. 'international nurses or overseas nurses or foreign nurses or migrant nurses' AND 'promotions in the workplace'

4. 'international nurses or overseas nurses or foreign nurses or migrant nurses' AND 'retention or attrition or turnover or intent to leave or intent to stay'

5. 'international nurses or overseas nurses or foreign nurses or migrant nurses' AND 'recruitment and selection'

6. 'international nurses or overseas nurses or foreign nurses or migrant nurses' AND 'career progression or promotion or career advancement or career development or career trajectories'

7. 'international nurses or overseas nurses or foreign nurses or migrant nurses' AND 'succession planning and leadership development'

8. 'international nurses or overseas nurses or foreign nurses or migrant nurses' AND 'attraction'

9. 'international nurses or overseas nurses or foreign nurses or migrant nurses' AND 'learning and development'

10. 'international nurses or overseas nurses or foreign nurses or migrant nurses' AND 'talent identification'

The most relevant three databases EBSCO, PubMed and Scopus were selected. In selecting these databases, the research team was guided by the advice of the librarians. These databases were perceived to be the most comprehensive for this topic of research, and in using these databases in our research, all the other databases were subsequently covered. Databases were accessed through the online portal of Glucksman Library.

The option of searching 'all my search terms' was selected on each database, using the search strategy outlined previously. The same search strategy was employed across all databases, to ensure that articles that one of the search terms was eliminated, for example articles pertaining to international nurses but not talent management. The total number of search results obtained from the three selected databases is shown in the Table 1 below.

## 3.4 Study selection

The search results on EBSCO, PubMed and Scopus databases were carefully screened by limiting the results to English only, and by selecting results from the last ten years only (2012–2022). Following are the inclusion criteria for the selected studies:

1. Only empirical studies, peer-reviewed studies and journal publications were included.

2. Studies were included only if they were published in the last ten years between 2012 and 2022.

3. Studies were included only if they were in the English language. We understand that relying on English-language studies may not represent all the evidence as it can create a language bias, however recent studies have shown minimal effect on the effect estimates and the overall conclusions of systematic reviews [22].

4. Studies were included only if they related to international nurses or overseas nurses or foreign nurses or migrant nurses as a specific cohort.

   Studies were excluded if they met the following criteria:

1. Conference proceeding papers, general magazine articles, commentaries, opinion pieces, newsletters, and any grey or secondary literature were excluded.

2. Studies were excluded if they were not in the English language.

3. Studies that were older than 2012 were not included.

4. Studies that were not about international nurses or overseas nurses or foreign nurses or migrant nurses were excluded.

5. Studies related to nursing students and their experiences were not included.

6. Opinion pieces and review studies were excluded from the final selection of studies.

## 3.5 Quality appraisal

Quality appraisals were first performed by reviewer one (N.R.) and was subsequently checked by the second reviewer (S.H.Z). Where queries or concerns arose, consensus was achieved

**Table 1. Total number of search results obtained from selected databases.**

| Sr No. | Database | Total Results | Date of Search |
|---|---|---|---|
| 1 | PubMed | 437 | 25/10/2022 |
| 2 | EBSCO | 517 | 28/10/2022 |
| 3 | Scopus | 400 | 02/11/2022 |

between the two reviewers. Following on from this, the identified search strategies were applied to the three databases and a list of studies totalling 1354 studies was generated. The search results were exported into EndNote software. The screening process for the articles began and initial title screening was conducted by two independent reviewers (S.H.Z., N.R.).

This stepwise process of title screening and abstract screening was organised in the End-Note software by maintaining records of included results and excluded results in separate folders. These EndNote folders were accessible by all the research team members. Once the abstract screening was completed, the process of full article screening began as the next step. The full article screening was also conducted by two independent reviewers (S.H.Z., N.R.). The final step of full article screening yielded 62 results to be included in the review.

### 3.6 Search outcome

Initially, 417 articles were eliminated from the total 1354 results from all databases, as they were irrelevant to the paper topic, or they did not meet the basic inclusion criteria of the study. The remaining 937 results were title screened for relevance to the topic and as a result 580 articles were removed; leaving 357 relevant results after title screening and 16 duplicate results were eliminated. The title screening process was followed by the next phase of abstract screening which was also conducted by two reviewers (S.H.Z., N.R.) independently. A total of 210 articles were excluded after the abstract screening and only 131 articles were remaining. 31 articles were excluded after the full article screening which resulted in 100 articles to be included. Out of those 100 results, 3 articles were removed because they were inaccessible for the researchers. The library was contacted, and the journals were contacted as well but these 3 articles could not be obtained, and this brought down the number to 97 articles. 35 articles were further removed as they were opinion pieces and review studies and did not have empirical evidence. As a result, the total number of results included in the final review came to 62 articles.

PRISMA Search Reporting Extension Checklist (PRISMA-S) was used to facilitate the reporting of this systematic literature review. The PRISMA flow chart process for this systematic literature review is diagrammatically represented in Fig 1. This diagram demonstrates the process flow of conducting the systematic literature review and the four main stages of conducting the systematic review which are identification, screening, eligibility, and inclusion.

### 3.7 Syntheses

Qualitative methodologies are used in a healthcare setting to try and make sense of the multiple complex processes [23]. Thematic analysis was used to uncover patterns in information or accounts of experience [24]. This type of qualitative data analysis is widely utilized but rarely recognised as a valuable and adaptable way for analysing qualitative data [25]. To provide rigour in thematic analysis it is important to follow a clear set of stages in the data analysis [26]. The research team followed the steps outlined by the researchers in [25] which include familiarisation with the data, generating initial codes/ themes, reviewing, and naming themes and finally producing the report [25].

In the process of reviewing the abstracts of the studies and applying inclusion and exclusion criteria to the search results, the authors (S.H.Z., N.R.) familiarized themselves with the broad themes and common patterns in the studies. The articles that did not address the talent management of international nurses in healthcare settings were omitted, and the study's inclusion and exclusion criteria were applied strictly. The same data familiarization process continued while the authors (S.H.Z., N.R.) were conducting the full screening of articles. The common patterns and themes emerging from the selected articles were highlighted and discussed in

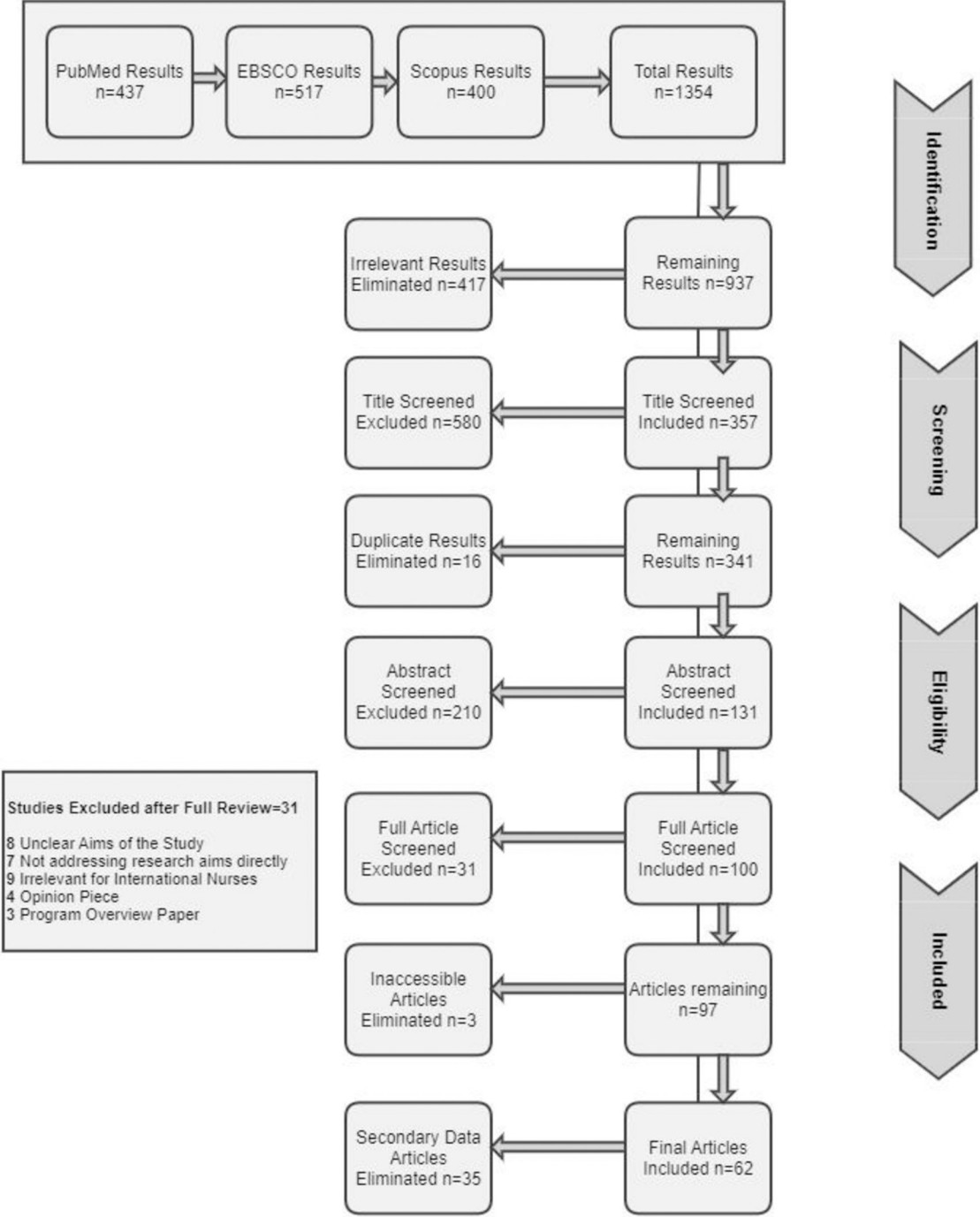

**Fig 1. PRISMA flow chart of systematic literature review for talent management of international nurses.**

depth between the authors (S.H.Z., N.R.) and the notes were matched regularly. Preliminary themes were developed by both authors (S.H.Z., N.R.) and were shared with the rest of the authors for discussion (E.B., H.P., C.OD., B.OM.).

Initially the themes of 'cultural appropriation' and 'communication' were developed separately and the most relevant corresponding articles were placed under these themes. However, as the process of full article screening continued, the authors noted the inter-linked

relationship of 'cultural appropriation' and 'communication' emerging in the relevant papers and how they shaped the talent management of international nurses in healthcare settings. As a result, it was decided to combine these two themes into one theme and rename it as 'culture and communication'.

Similarly, with the full reading of selected articles, a new theme emerged for 'racism and discrimination' which was not envisioned as a theme at the initial stage of the analysis, but as the in-depth article review and analysis process continued, this theme emerged from within 8 articles which is a considerable number, therefore, the theme of 'racism and discrimination' was added to the final list of themes. The final themes were once again discussed among all authors and a consensus was reached before conducting the full-text review of articles under each thematic code.

The final 62 included articles were carefully placed under the corresponding themes by the authors (S.H.Z., N.R.) and each team member (S.H.Z., N.R., E.B., H.P., C.OD., B.OM.) picked a theme to work on, which included the full-text review of all articles that were placed under the respective theme. This way, all the team members (S.H.Z., N.R., E.B., H.P., C.OD., B.OM.) took part in conducting the thematic analysis of the final articles; thereby ensuring the rigour of the synthesis.

The articles that focused on the hiring, recruitment and selection of international nurses were grouped together under the theme of 'recruitment and selection' and this theme had one of the highest number of articles in it. The articles that were related to the topics of promotion and career progression of international nurses were grouped under the theme of 'career progression and promotion'. Similarly, the articles related to the topics of learning, skill-building and professional development of international nurses were grouped under the theme of 'learning and professional development'. The articles that emphasized on international nurses' intention to leave, intention to stay, or discussed the problems of this cohort around attrition, retention and turnover were grouped together under 'retention and turnover'. Furthermore, in some articles, the challenges of discrimination, prejudice, racism, and inequalities experienced by international nurses in their working environments were dominant, therefore, such articles were combined under the theme of 'racism and discrimination'. The largest pool of articles was focused on the cultural integration, communication challenges and language barriers in the host country and how they impact the working experiences of international nurses, consequently shaping their talent management; and total 19 articles were grouped together into the theme of 'culture and communication'.

Therefore, based on the commonalities in the reviewed articles, the final 62 articles were thematically coded in an Excel sheet under the following six thematic categories:

- recruitment and selection (14 articles),

- career progression and promotion (4 articles),

- learning and professional development (9 articles),

- retention and turnover (8 articles),

- racism and discrimination (8 articles),

- culture and communication (19 articles).

## 3.8 Data abstraction

Once the themes were decided, the final articles were divided among six independent reviewers from the research team (S.H.Z., N.R., E.B., H.P., C.OD., B.OM.). Each reviewer worked on

their share of articles by reading through the full articles and summarizing the key findings under a preset template for data extraction drawn upon the study protocol with some additional information items.

## 4. Results

### 4.1 Study characteristics

In the review, there were 32 qualitative studies, 24 quantitative studies and 6 mixed method studies. The studies included in the review were diverse in terms of the country contexts, where 19 studies were from the European region, 13 were from the United States, 4 were from Canada, 5 were from Australia, 5 were from Saudi Arabia, 2 were from Asia, 4 were from New Zealand, and 6 were multiple country studies. Most of the studies were from high-ranking journals, i.e., 47 studies were from Q1 journals, 8 studies were from Q2 journals, and the rest were from Q3 journals.

### 4.2 Risk of bias

Several steps were taken throughout this systematic review to minimise risk of bias. To start, guidance was sought from librarians in the Glucksman library when selecting the search terms, and relevant databases. Regular meetings were held between the two reviewers (S.H.Z., N.R.) who collated data at each stage of the process and discussed any ambiguities, ensuring clarity and transparency throughout the process. Three other independent reviewers joined the review process at the stage of final paper review (S.H.Z., N.R., E.B., H.P., C.OD., B.OM.).

The results from all databases were exported into the EndNote software and organised into dedicated folders. For transparency, access to these folders was granted to all six research team members (S.H.Z., N.R., E.B., H.P., C.OD., B.OM.), and the review process was then completed by the team (S.H.Z., N.R., E.B., H.P., C.OD., B.OM.) who analysed each article independently and summarized the findings in a data extraction template.

Each study included in this systematic review was carefully assessed by two independent reviewers (S.H.Z., N.R.) for its quality assurance and methodological rigour. The two reviewers (S.H.Z., N.R.) regularly discussed the 'risk to rigour' for the selected studies [27] and the methodological limitations of the individual studies were discussed to determine whether they would impact the findings of our systematic review. Studies were excluded if they did not have clear aims and objectives, or relevant research question focused on aspects of talent management of international nurses in healthcare settings. Moreover, studies were only included in the final review if they provided empirical evidence through primary data and had rigour of participants' identification, sampling, and data collection to address the research questions, and were excluded if they were opinion pieces, lacking academic rigour and precision. During this process, it was ensured that the exclusion of studies was not entirely based upon the inclusion and exclusion criteria; rather the screening process also gave due consideration to the quality assurance of the studies.

### 4.3 Limitations of the study

The findings of our systematic literature review should be interpreted with the following caveats in mind:

1. Only papers written in English were included in this study, we note that this may lead to a lack of generalizations in our findings.

2. There is a lack of similar studies in past literature which is a limitation. Although there have been studies about staff retention, turnover and motivation to stay, etc. but there has been little focus on talent management of international nurses overall.

3. This systematic review only included primary studies, and therefore data from secondary studies or grey literature and government sources were not included. The decision to remove secondary studies and grey literature was to ensure the review was a synthesis of the existing empirical data on talent management of international nurses.

4. This study only included peer-reviewed studies from the last ten years.

## 5. Discussion

The in-depth analysis of the reviewed articles was conducted under different thematic categories of talent management as explained earlier. The findings of this thematic analysis are narrated in separate themes below, followed by an overall discussion.

### 5.1 Findings

**5.1.1 Recruitment and selection.** While the recruitment of international nurses is an effective means of reducing staffing shortages in the host countries, the literature highlights several challenges, which are first experienced at the recruitment stage of the process. Data from the US showed that participants from low-income countries were at a higher risk of experiencing poor recruitment practices than workers from high income countries [28]. Findings from the research [29] analysed the recruitment drive by NHS Trusts in England for hiring international nurses from non-EU countries instead of hiring nurses from within the EU countries. Another study [30] examined the heavy reliance on international nurses to fill healthcare vacancies in Ireland over a decade (from 2000 to 2010) as a temporary solution or a quick fix. This recruitment drive failed to address the underlying causes of staff shortages in Ireland as it lacked long-term perspective and strategic workforce planning. This short-term solution to fill vacancies in the Irish healthcare system without any strategic planning is accompanied by many challenges; thereby emphasizing on the need to have an active international workforce recruitment agenda embedded within the comprehensive health workforce planning as a way forward.

Similarly, a study [31] within the NHS found that nurses recruited from Europe were less likely to tolerate unfavourable recruitment processes, while their non-European counterparts tolerated such unfavourable practices due to factors such as obligation to send money back to their family in their originating country despite the emotional toll of working within trying working conditions. Researchers have elaborated the transition and adaptation challenges faced by OQI (Overseas Qualified Nurses) working in Australia and the gaps between expectations and the realities [32]. These international nurses experience many lengthy bureaucratic obstacles during migration, for example the process of migrating to Australia is extensive including the need to establish identity, meet and pass the English language proficiency tests, and meet Australian nursing education standards, which can take up to two years [32]. Migration can also be a costly process for international nurses. Within the Canadian context, researchers found that international nurses used savings or received financial assistance from family members to recertify to work in Canada [33]. Furthermore, once working within the host system, international nurse is more likely to experience discriminatory behaviour, for example, lower levels of pay and benefits, less favourable shifts, such as night shifts or evening shifts, and be recruited to units with roles that are often difficult to fill [34]. Research has

highlighted the inequalities experienced by Filipino nurses working in Finland; especially their lack of Finnish language knowledge being used as an excuse for their low salaries [35].

International nurses also note the lack of orientation upon arrival. Researchers examined the experiences of international nurses educated in EU countries working in Norway and reported that better orientation programs and workplace integration into the local healthcare systems are required [36]. Literature highlights the need for more thorough orientation and training programs involving language proficiency and regional healthcare systems [37]. On a positive note, a study [38] found that a high number of respondents, in the US context, felt that international nurses are as qualified as US nurses and found that nurses educated aboard can be of benefit to the team. Finally, it was found that the international experienced gained by international nurses meant that this cohort of nurses felt they would be in a better position to secure roles in the future because of the confidence gained from working in other cultures, from a social cultural perspective [39].

Studies also highlight the remittance practices for international nurses, where these nurses send funds home to contribute to the safety and well-being of their family. In fact, these remittance practices of healthcare workers employed in another country can account for a substantial amount of their home country's GDP [40]. The studies suggest that because of these practices pay is seen as a more important factor than individual comfort and responsibility [41] and the obligation to comply with this cultural expectation results in the acceptance of unfavourable recruitment process and working conditions [31] which disproportionately, negatively effects international nurses from lower income countries [28]. To compound this several studies highlighted the exorbitant contractual breech fees, threats of visa and green card cancellation, as well as legal action for immigration fraud were made against international nurses who attempted to quit their jobs [42]. Though these international nurses are receiving high wages in the host countries, their social status is decreased reinforcing global capitalism with increased racial and gender inequalities for this cohort [43].

**5.1.2 Career progression.** Career progression, or lack thereof, for international nurses is an area of concern within the nursing profession. Researchers highlight lack of career choices and professional development opportunities for international nurses which subsequently causes frustration and lack of job satisfaction [44]. Furthermore, they suggest that a key area of concern is the mismatch between the job expectations of international nurses and the positions that are made available to them upon arrival [44]. International nurses are often placed in positions that do not match their qualifications and experiences, meaning international nurses are placed in positions below that of their qualifications and as a result health care organizations are not maximizing this cohort of nurses to their full potential. Evidence to date also suggests that international nurses progress at a slower pace than their domestically trained counterparts, and international nurses are less likely to participate in professional development [45].

Researchers emphasize on the importance of career development needs of international critical care practitioners; including international nurses [46]. Within the literature mentorship is highlighted as being critical for the career advancement of nurses [45, 47, 48], however the level and quality of mentorship received by international nurses is insufficient in aiding them to advance to positions of leadership, unlike their domestically trained counterparts. In terms of career progression within the US context, it is found that two out of every eight internationally trained nurses compared had been promoted over the past three years, compared to two out of every five US trained nurses [45]. Furthermore, a study found that few international nurses reported positive career development post migration [49]. In terms of the impact migration had on the individuals' career, almost all the respondents experienced short term, long term or permanent inability to work as healthcare professionals. Many were unable to

work according to their qualifications due to the fact that their diplomas are not recognized in the host country, while many had to change career paths.

**5.1.3 Professional development.** An attractive incentive for nurses to travel is their potential to undertake a bridging program in their new host country [50, 51] aligning with some internationally educated nurses' level of qualifications which require 'top up' qualifications thus, supporting their professional development. Nurses' own acknowledgement of their professional skills and competencies are key in ensuring that they can practice in a safe way; and this is specifically important in preparing nurses for the demanding process of migration [52]. The process of moving and travelling to support professional development is challenged when difficulties arise in the nurse registration process in the host country [53]. In recognising this, the Canadian regulator developed, implemented and evaluated changes to internationally educated nurses' registration policies and practices at the College and Association of Registered Nurses of Alberta with positive results. Having an efficient and transparent regulatory policy informed by evidence, streamlined the registration process resulting in positive experiences for applicants with reduced waiting periods for approval [53]. This is an action that could be replicated in other jurisdictions supporting nurses' professional development while also reinforcing recruitment strategies for healthcare organisations.

A study suggests that international nurses are encouraged to engage in governance and professional development opportunities to support their transition to work in a new country however, they did not value participation in governance and experienced poor professional development as they lacked guidance on the process for career progression [54]. Most international nurses make the transition and adaptation to their new country however, African nurses were reported to find this challenge greater because of their reduced ability to engage in professional development opportunities [55]. Thus, as a result of lack of engagement in professional development this reduced job opportunities and also reduced their ability to transition to their new country and home. Research suggests that family and caring responsibilities towards children and elders had huge implications for the careers of Asian nurses working in New Zealand; especially their abilities to continue education, participate in opportunities of professional development ultimately affecting their intention to stay in their jobs in New Zealand in the longer run [56].

Professional supports such as formal mentoring programs aid and strengthen transitions [57]. Lowering levels of moral distress have also been reported to contribute to decreased turnover and stabilization of the critical care workforce [57]. Moral distress is an important reason for international nurse's intentions to leave their jobs therefore, providing supports such as tools and strategies for nurses by management and the organisation demonstrates their support in trying to help reduce staff turnover. Positively, international nurses in a study [58] found when working with local nurses, that being respected by seniors and management along with the understanding that working abroad develops their career, supported an increased level of job satisfaction. These areas are points for consideration by senior management in supporting the professional development and recruitment and retention of internationally trained nurses. Researchers elaborated the importance of developing global nursing leadership for international nurses to bring about positive changes in nursing practice especially in empowering them and extending opportunities of capacity building [57]. Although professional development for nurses was discussed in some papers in this review, it is noteworthy that its absence in some of the workplaces for nurses is not a leading factor in their decisions to leave their job.

**5.1.4 Retention and turnover.** Retention and turnover are critical components of the talent management process with each of these topics widely considered in academic research [59, 60]. Eight articles identified in our literature review were published on the topics of retention

and turnover intent, providing evidence on the antecedents to these outcomes for international nurses. Though the findings from the studies presented were based on a sample that constituted international, foreign trained immigrant nurses; several studies considered the factors measured in comparison to home country nurses and from their findings suggest that though there are similarities, there are also important different employment issues influencing the retention and turnover of IENs (Internationally Educated Nurses) which must be considered [14, 61, 62].

A study examined the challenges of anticipated nursing workforce shortage in the UK since Brexit which has imposed immigration and employment restrictions on EU nurses in the UK [63]. A consistent message across multiple studies was the effect of negative human resource practices by host countries which had a detrimental effect on retention and turnover. Geun and colleagues highlighted the importance of poor communication practices as part of the recruitment process resulting in pre and post expectation gaps [41]. Through their research they examined predictors of turnover among Asian foreign-educated nurses (FENs) in the United States in their first year of employment. This study highlights the importance of the expectation gap of Asian Foreign trained nurses pre and post migration and reports that on average pre-migratory expectations were higher than post-migratory experiences and that FENs that reported lower organisational responsibility were more likely to leave in their first place of employment. This sample also reported a higher gap for reward than for comfort and responsibility which could be explained by the cultural requirement to send remittance to their family [41]. Researchers have addressed the factors that influence migrant nurse turnover in the MOH hospitals of Saudi Arabia, identified nine categories that effected turnover including professional growth and development, leadership style, management, wage and benefits, workload, interpersonal relationships, housing facilities and services, hospital facilities and intent to stay and turn-over intention [64]. Of these categories wage benefits and workload factors were the most important causes of turnover, followed by inadequate housing and hospital facilities. Similarly, researchers have found workload factors affected turnover intent as well as other push factors including poor salaries, working conditions and resource shortages, job uncertainty and lack of investment in the healthcare system [65]. Foreign nurses employed in Saudi that it was a younger group of healthcare workers who tended to leave their jobs whereas those with 5–6 years clinical experience and no experience abroad had a significantly lower level of intention to leave [14]. Researchers have studied the antecedents for job satisfaction in the form of life satisfaction, self-esteem, and perceived stress among immigrant Korean nurses working in the U.S., suggesting the importance for administrators and home country nurses to understand and develop programs around these topics as a way of increasing job satisfaction which in turn will positively affect retention and reducing turnover [61].

The importance of age and culturally appropriate retention strategies are considered in a study which was designed to understand the experiences of nurses aged under 30 in the New Zealand workforce with a view to developing age-appropriate retention strategies [62]. They found that for young nurses and internationally trained nurses that are categorised by the Nursing Council of New Zealand as Asian [62] were five important themes to consider: nursing problems, nursing benefits, being young, coping, and generational disparities. For international nurses they found that younger nurses in New Zealand have additional obstacles if they are Asian. They struggled to reconcile their viewpoint of being taught to give a level of nursing care that may demand them to question their elders. They felt culturally obligated to do what older Asian nurses and patients instructed them without inquiry. The younger Asian nurses were aware of how unjust this was and how it affected their ability to do their duties which effects retention and turnover. The importance of effective orientation programs for FENs was considered by Geun and colleagues when studying Asian nurses from the Philippines and

Korea working in the United States. Their study found that 'perceived quality of orientation predicted organizational-level turnover and trended toward predicting unit-level turnover' [66, p. 519].

**5.1.5 Discrimination and racism.** Literature suggests that discrimination and racism are important factors in determining international nurses' retention and intention to leave [54, 67]. A study was conducted on the experiences of migrant nurses in the United Kingdom and found that racial microaggressions were often experienced by these nurses from colleagues and patients alike [68]. It was discussed that patients in the host country had certain racial preferences for nurses, and racist bullying from colleagues. Such subtle incidences of racial microaggressions led to feelings of exclusion, isolation, feeling unwanted, anger, and negatively impacted the emotional health of the migrant nurses. Similarly, institutional racism was experienced by this cohort of nurses which created barriers for their access to opportunities of further training and promotion; however, these incidents were often diminished because of their subtle nature. Researchers [69] found that internationally qualified nurses experienced more discrimination and racism in hospitals in the United States. The prejudice and differential treatment towards foreign-qualified nurses were projected by fellow nurses, supervisory hospital staff and patients. Interestingly, it was noted that mistrust in the ability of nurses to provide quality patient care was based on their skin colour or nationality and although locally educated nurses experienced discrimination as well, it was more pronounced for foreign-qualified nurses [69].

Research found that poor working conditions, insufficient professional support, excessive workload and difficult living experiences were the main reasons behind the migrant nurses' choice of not working in primary care in their home countries [70]. Therefore, these nurses prefer to migrate to other countries seeking better professional opportunities and improved working conditions. A study elaborated on the problems of alternative recruitment and migration systems for international nurses in the UK; despite stringent immigration policies which leads to increased exploitation and vulnerable working conditions for these nurses [71]. They discuss the experiences of racism and bullying experienced by Malawian and Nepali nurses working in the UK that were projected by their managers and colleagues and how this may lead to the feelings of isolation for these international nurses. Research has narrated the experiences of Chinese nurses working in New Zealand during Covid-19 and the results showed racial discrimination and workplace bullying against Chinese nurses by their colleagues [72]. Since the virus was commonly associated with China, these nurses felt isolated and helpless due to racist remarks made in this context.

Researchers examined the retention challenges for Asian nurses working in Saudi Arabia and suggest that unfair renumeration for the nursing jobs can negatively impact the intention to stay [67]. Variable renumeration was reported in the study on the basis of nationality, race and migration status, thus calling for fair renumeration policies on the basis of professional competence. It was discussed that social support from the immediate supervisor, organisational commitment and autonomy can substantially decrease international nurses' turnover [67]. The same was reported for Chinese nurses in New Zealand by researchers [72] who received positive workplace support from their colleagues and recognition by the public in New Zealand during the pandemic.

It is discussed that international nurses are likely to experience discriminatory behaviour in the host country; like poor pay and benefits, excessive workload, night shifts, placed in departments with challenging tasks [73]. Research has elaborated that international nurses working in Saudi Arabia had lower job satisfaction as compared to the local Saudi nurses as they were not satisfied about their extrinsic rewards like salaries, holiday entitlement neither the overall work-life balance [74]. These migrant nurses in Saudi Arabia reported tack of competitive

pension resulting in poor future security; impacting their intention to stay in the country long term. Almansour and colleagues emphasized the need to consider nationality as an important factor in determining job satisfaction of the international workforce.

**5.1.6 Culture and communication.** The challenges of cultural integration and communication are mostly underestimated [75] but they have significant implications on shaping the working experience of international nurses [75–77] and in determining their job satisfaction [78, 79]; thereby underpinning the importance of effective talent management. The migration of international nurses comes with unique challenges as emphasized in nursing literature and calls for careful considerations by healthcare organisations, administrators and policymakers to ensure smooth transition into the new culture [77, 80–82].

Researchers have identified cultural differences, communication barriers and variable nursing practices in the home country as the key determinants of international nurses' adaptation and integration into the new healthcare settings [75, 80, 82, 83]. Among other factors like salary and associated benefits [78, 81], supportive work environment [77, 84, 85], social networks [85] language proficiency was noted as the major aspect for the job satisfaction among international nurses [75, 78, 80–82, 86]. Language proficiency is indispensable for international nurses to effectively provide caring services to the patients and in building interpersonal relationships [75, 81]. At the same time, language plays a key role to comfortably work in multicultural teams and integrate into the new workplace settings [87]. They presented the case of international nurses working in New Zealand and argued that communication gaps between host country nurses and international nurses posed threats to efficient teamwork and risked patient safety [87] and brought interpersonal conflict between host country nurses and international nurses [79].

A study [82] found that international nurses from non-English speaking countries struggled with communication barriers; and were subject to discrimination and racism in the Australian context as the host country. Similar challenges of cultural integration were reported in a study [81] for Filipino nurses in the US context and in another study [88] for Indian nurses working in Italy. Research has highlighted similar factors like English language barrier, challenges of social and professional adjustments as the main concerns for the integration of international nurses in China [86]. This evidence is sufficient to highlight the importance of providing adequate training and orientation programs on communication and cultural integration [76, 83, 86, 89] cross-cultural competence [90] language and communication training [78, 83, 84] and streamlining systems of nursing credentials [84] for incoming international nurses.

The need for UK organisations to consider cultural integration needs of international nurses was discussed by researchers [83] as they presented the case of Indian nurses' recruitment in the NHS and the introduction of bespoke mentorship program on language, cultural integration and skill-based training. Research has highlighted the benefits of training international nurses on intercultural communication [89]. The benefits of a bridging program for international nurses in Sweden was narrated in another study [91] although it did not address all the challenges of cultural integration, but it was useful to improve language skills and nursing skills of this cohort. Researchers have attributed the smooth experience of migration of international nurses to the constant interaction and adjustment between nurses and organisations which presents a holistic approach to recruiting and talent management of international nurses where organisations share the responsibility of professional transitioning and cultural integration [85]. The unmet expectations of international nurses in Germany result in poor cultural integration and low retention as a result [92].

Literature examines the historical stigma associated with female nursing professionals from India; thereby, the discrimination and anti-immigrant rhetoric in Italy may cause hurdles in the social and professional integration of Indian nurses [88]. Studies [81, 93] shed light on the stress associated with migration, since the struggle is manifold for international nurses, as

compared to the local nurses; such that they must resort to appropriate coping strategies to address the physical and psychological impact of migration-related stress [81] also presented in the case of Polish nurses working in the UK [93]. A study [92] has reported lower workforce burnout and better work-life balance among international nurses with higher language proficiency. Nonetheless, researchers [90] emphasize on the positive relationship between cross-cultural competence and wellbeing of both native and international nurses working in Finland. They present cross-cultural empathy to have great benefits for nurses' health and wellbeing; especially on perceived time pressure, distress and sleep problems [90].

## 5.2 Conclusion

Talent management in nursing to date remains an understudied phenomenon and an area worthy of exploration due to the progressive international crisis in nurse recruitment and retention. More focused again within talent management is the area of supporting the recruitment and retention of international nurses. As a phenomenon, talent management is universally experienced across all care sectors and one which requires a body of research to build support for nurse recruitment and retention. This systematic review is timely to present and explain the challenges of talent management for international nurses; thus, providing rich data under the different thematic categories of talent management; i.e., recruitment and selection, career progression, professional development, retention and turnover, discrimination and racism, and culture and communication. There is scarcity of available academic literature on the talent management of international nurses to inform practice, therefore, there is a need to conduct rigorous empirical research in this area. Although literature acknowledges the processes of talent management independently, it pulls away from categorically acknowledging talent management as a holistic approach. Talent management of international nurses needs to be carefully strategized and thoroughly incorporated into the healthcare workforce planning to bridge the gaps between expectations and realities of migration. Lengthy and costly nurses' registration processes, excessive workload and little financial benefits, poor work-life balance, low job satisfaction, communication barriers and cultural integration challenges, racial discrimination and unequal professional opportunities in the host country are some of the main factors that negatively impact the retention of international nurses. The onus of migration and transition into the host country lies mostly with the individual nurse, and there is little financial and operational support made available from the organisations. With transparent and rigorous processes of talent management of international nurses along with improved working conditions, attractive financial compensation packages, smoother cultural and workplace integration, mentoring and career development programs, healthcare organisations would be better equipped to retain international nurses and reduce turnover. It can be valuable to build cross-cultural competence of the host country staff along with incoming international nurses. Introducing training around diversity and inclusion and intercultural competence as part of talent management would be greatly beneficial for the retention of international nurses. It is also important to acknowledge that the studies included in this systematic literature review do not consider the implications of Covid-19 on the migration of international nurses. As a future research direction, it would be useful to conduct a sensitivity analysis in the context of Covid-19 to determine how it has affected the global migration of international nurses and their talent management in healthcare settings worldwide.

## Supporting information

**S1 Checklist.**
(DOC)

**S1 Data.**
(XLSX)

## Author Contributions

**Conceptualization:** Sidra Hareem Zulfiqar, Nuala Ryan, Elaine Berkery, Claire Odonnell, Helen Purtil, Bernadette O'Malley.

**Data curation:** Sidra Hareem Zulfiqar, Nuala Ryan.

**Formal analysis:** Sidra Hareem Zulfiqar, Nuala Ryan, Elaine Berkery, Claire Odonnell, Helen Purtil, Bernadette O'Malley.

**Funding acquisition:** Nuala Ryan, Elaine Berkery.

**Investigation:** Sidra Hareem Zulfiqar, Nuala Ryan.

**Methodology:** Sidra Hareem Zulfiqar, Nuala Ryan.

**Writing – original draft:** Sidra Hareem Zulfiqar, Nuala Ryan, Elaine Berkery, Claire Odonnell, Helen Purtil, Bernadette O'Malley.

**Writing – review & editing:** Sidra Hareem Zulfiqar, Nuala Ryan, Elaine Berkery.

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
