## [Decision Letter · Decision Letter 0]

18 Aug 2023

PONE-D-23-21946Talent Management of International Nurses in Healthcare Settings: A Systematic ReviewPLOS ONE

Dear Dr. Zulfiqar,

Thank you for submitting your manuscript to PLOS ONE. After careful consideration, we feel that it has merit but does not fully meet PLOS ONE’s publication criteria as it currently stands. Therefore, we invite you to submit a revised version of the manuscript that addresses the points raised during the review process.

ACADEMIC EDITOR:

Dear authors,

The article needs a through revision by incorporating the comments of the esteemed reviewers. 

With regards,

Ranjit

We look forward to receiving your revised manuscript.

Kind regards,

Ranjit Kumar Dehury

Academic Editor

PLOS ONE

4. Please include your table as part of your main manuscript and remove the individual file. Please note that supplementary tables be uploaded as separate "supporting information" files.

Additional Editor Comments:

Dear authors,

The article needs a through revision by incorporating the comments of the esteemed reviewers.

With regards,

Ranjit

Reviewers' comments:

Reviewer's Responses to Questions

**Comments to the Author**

1. Is the manuscript technically sound, and do the data support the conclusions?

Reviewer #1: Yes

Reviewer #2: Yes

2. Has the statistical analysis been performed appropriately and rigorously? 

Reviewer #1: N/A

Reviewer #2: N/A

3. Have the authors made all data underlying the findings in their manuscript fully available?

Reviewer #1: Yes

Reviewer #2: No

4. Is the manuscript presented in an intelligible fashion and written in standard English?

Reviewer #1: Yes

Reviewer #2: Yes

5. Review Comments to the Author

Reviewer #1: Dear authors

the manuscript "Talent Management of International Nurses in Healthcare Settings: A Systematic Review" made it possible to identify the main themes associated with Talent Management of International Nurses in Healthcare Settings. It identified a series of factors that should be the subject of reflection, as it jeopardizes the human rights of nurses who migrate. The authors are to be congratulated for having emphasized and highlighted these aspects.

The introduction and background are adequate and justify the relevance of the systematic review. The objective is clear and adequate.

In the method, it is not clear how the quality assessment of the included articles was made. Has the review been registered with the Open Science Framework (OSF) or PROSPERO? It is not clear how the thematic analysis was carried out. In the results there is no description of the main themes (there is no analysis of the attached table), they are only presented in the discussion.

The limitations of the review are not presented.

Best Regards

Reviewer #2: Nice relevant review though not an new concept, it does require a detailed new perspective post pandemic scenario.

Some of my concerns are as follows.

1. Why only 2012 to 2022 years selected ? why not earlier? Also restricting to english language may also limit the review's generalization. This has to be discussed as a limitation.

2. Search terms on training, language and cultural competence, language barrier and economic status or financial reasons, support etc are not included and they have come as possible domains or themes in the end.

3. In this topic grey literature may be important and why is it excluded may be explained in methods or in discussion?

4. Sensitivity analysis with covid onset/ peak as a time point would also be useful post hoc analysis. Covid changed perspectives such as people wanting to be closer home and others capitalising on the increased demand for nurses all over the world.

5. How did the authors decide on thematic coding of 6 categories? How did these themes emerge ? Thematic coding by one reviewer can be biased. How did the other reviewers contribute to the coding? Did they concur fully or were there any differences ?

6. Risk of bias is also for individual study included in the study assessing for their quality and robustness of the results. Any standard assessment for appropriate study type may be used and reported. Risk of bias is not related to the "review process" but to the studies included. This section needs rewriting and reporting on quality of the included studies.

7. Emigration to some countries are easier for people from specific countries and that may show up as preferences but while really be only convenience and necessity.

8. With increasing risk of violence against heath workers and majority of nursing workforce being women, was there any themes on violence at workplace, harassment at workplace and harassment outside workplace be discussed in any of the include studies?

9. To refer to chapter and make changes as per recommendations and especially to correct section on risk of bias and to report same as per recommendations

https://training.cochrane.org/handbook/current/chapter-21

10. Articles excluded after full text review may be mentioned as a table with reasons for exclusion. These articles may serve as additional sources for the readers if they are interested in and provide a transparency to the review method too. ( May add this as a supplement)

6. PLOS authors have the option to publish the peer review history of their article (what does this mean?). If published, this will include your full peer review and any attached files.

Reviewer #1: **Yes: **Luís Manuel Mota de Sousa

Reviewer #2: **Yes: **Akilesh R

---

## [Author Response · Author response to Decision Letter 0]

2 Oct 2023

Dear Reviewers

Thank you very much for your time to review this paper. Your comments have been very useful in improving the quality of this paper. We have made all the required changes and the revised manuscript is attached herewith. The detailed response to reviewers letter is attached as well, where each comment is addressed separately. Thank you once again.

Best wishes

---

## [Decision Letter · Decision Letter 1]

20 Oct 2023

Talent Management of International Nurses in Healthcare Settings: A Systematic Review

PONE-D-23-21946R1

Dear Dr. Zulfiqar,

We’re pleased to inform you that your manuscript has been judged scientifically suitable for publication and will be formally accepted for publication once it meets all outstanding technical requirements.

Kind regards,

Ranjit Kumar Dehury

Academic Editor

PLOS ONE

Additional Editor Comments (optional):

Dear Authors

Based on my review and comments of esteemed referees, the article is found to be of publishable quality. Hence, it is accepted.

With regards,

Ranjit

Reviewers' comments:

Reviewer's Responses to Questions

**Comments to the Author**

1. If the authors have adequately addressed your comments raised in a previous round of review and you feel that this manuscript is now acceptable for publication, you may indicate that here to bypass the “Comments to the Author” section, enter your conflict of interest statement in the “Confidential to Editor” section, and submit your "Accept" recommendation.

Reviewer #1: All comments have been addressed

Reviewer #2: All comments have been addressed

2. Is the manuscript technically sound, and do the data support the conclusions?

Reviewer #1: Yes

Reviewer #2: Yes

3. Has the statistical analysis been performed appropriately and rigorously? 

Reviewer #1: N/A

Reviewer #2: N/A

4. Have the authors made all data underlying the findings in their manuscript fully available?

Reviewer #1: Yes

Reviewer #2: No

5. Is the manuscript presented in an intelligible fashion and written in standard English?

Reviewer #1: Yes

Reviewer #2: Yes

6. Review Comments to the Author

Reviewer #1: Dear authors

The manuscript "Talent Management of International Nurses in Healthcare Settings: A Systematic Review" was carefully revised and the quality was improved.

There was concern about responding appropriately to reviewers' comments.

However, there was no response regarding registration with PROSPERO.

It is recommended to place the limitations of the review at the end of the discussion.

Best Regards

Reviewer #2: The article has addressed the comments in a suitable way and made the necessary changes been made in the submission.

7. PLOS authors have the option to publish the peer review history of their article (what does this mean?). If published, this will include your full peer review and any attached files.

Reviewer #1: **Yes: **Luís Sousa

Reviewer #2: **Yes: **Akilesh R

---

## [Editor Report · Acceptance letter]

27 Oct 2023

PONE-D-23-21946R1 

Talent Management of International Nurses in Healthcare Settings: A Systematic Review 

Dear Dr. Zulfiqar:

I'm pleased to inform you that your manuscript has been deemed suitable for publication in PLOS ONE. Congratulations! Your manuscript is now with our production department. 

Kind regards, 

on behalf of

Dr. Ranjit Kumar Dehury 

Academic Editor

PLOS ONE